# The Effect of Cu Addition on the Precipitation Sequence in the Al-Si-Mg-Cr Alloy

**DOI:** 10.3390/ma15228221

**Published:** 2022-11-19

**Authors:** Bin Chen, Liang Dong, Bin Hu, Zhenyang Liu

**Affiliations:** 1School of Materials Science and Engineering, Shanghai Jiao Tong University, Shanghai 200240, China; 2China Science Laboratory, General Motors Global Research and Development, Shanghai 201206, China

**Keywords:** Al-Mg-Si-Cr alloy, Cu addition, precipitation, GP zone, HAADF-STEM

## Abstract

In this work, the effect of Cu additives and heat treatment on the precipitation sequence of an Al-Si-Mg-Cr alloy has been systematically studied by means of advanced spherical aberration-corrected electron microscopy. Cu atoms tend to gather at the interface between the precipitates and the matrix at the beginning of the aging process. Then, Cu atoms diffuse into the precipitates. Two types of GP zones are formed in the first stage of precipitation: one is the type I GP zone and the other is the type II GP zone. The type I GP zone $\beta_{\mathrm{Cu}}^{\prime \prime }$ evolved into the Q′ phase, while the type II GP zone evolved into the θ′ phase during the aging process. The aging sequence of the Al-Si-Mg-Cr alloy can be determined as a supersaturated solid solution (SSSS) → GP zones → β″→ β′/B′(→β). The aging sequence of the Al-7%Si-0.3%Mg-0.3%Cr-1.5%Cu alloy can be determined as a supersaturated solid solution (SSSS)→GP zone→$\beta_{\mathrm{Cu}}^{\prime \prime }$→Q′ + θ′(→Q + θ).

## 1. Introduction

Al-Si cast alloys are currently the most widely used commercial cast aluminum alloys in the industry due to their excellent combination of good castability, high strength/weight ratio, impressive mechanical behavior, and superior corrosion resistance. To date, microalloying and heat treatment have been effective methods for improving the mechanical properties of Al alloys.

Over the years, a small amount of Mg has been added to Al-Si cast alloys, with the aim of obtaining improved mechanical performance via solution heat treatment and artificial aging [1,2]. The strength of the alloy can be effectively improved by uniformly distributed nanoscale Mg-Si-rich precipitates formed during aging. These precipitates include β-Mg_2_Si and its precursors, β″ and β′, which can act as strong obstacles to the movement of dislocations and grain boundaries during the deformation of Al-Si alloys and thus increase their mechanical strength. In addition, the coexistence of clusters and precipitates in the aged Al-Mg-Si alloys has been studied by Shishido et al. [3] to investigate their contributions to the strength of the alloys.

In recent years, eliminating the detrimental effect of impurity elements, such as Fe, which diminishes the microstructure and mechanical properties of Al alloys through the microalloying method, has received considerable attention [4,5,6,7,8,9,10]. Our early work found that Cr additives greatly reduce the negative effect of Fe by transforming the Fe-containing impurity phase to finer Cr-Fe-rich dispersoids, which are fully coherent within the surrounding matrix [8]. In addition, these uniformly distributed dispersoids with a high density can effectively pin the movement of dislocations and bring dispersion-strengthening effects to Al alloys [9].

Cu, as an important alloying element in the Al alloy system, is promising for remarkably enhancing mechanical performance through the heat treatment processes [11,12]. As Li et al. [12] reported, through a certain heat treatment consisting of solution treatment and artificial aging, the strength of Al-Si cast alloys can be increased considerably. In recent years, it has been reported that adding Cu to Al-Mg-Si alloys can enhance the age-hardening effect of the alloy [13,14,15,16]. Sunde et al. [15] found that relatively small changes in the Cu level and the Si:Mg ratio had a significant effect on the resulting distribution of precipitate phases, their structural evolution, and their thermal stability. The addition of Cu promotes the phase transition from β-Mg_2_Si to Q′ [17,18]. Matsuda [19] et al. also reported that Cu atoms tend to segregate at the interface between Q′ and the matrix, which can effectively reduce the free energy of the system, inhibit the growth of precipitates, and refine the microstructure of the alloy. The findings of Tavitas-Medrano et al. [20] show that the main strengthening phase in cast Al-Si-Cu-Mg alloys is θ-Al_2_Cu.

In this work, Cu will be added to an Al-Si-Mg-Cr cast alloy to study its influence on the properties and microstructures, especially the precipitation sequence. The microstructure evolution in both the Al-Si-Mg-Cr alloy and the Al-Si-Mg-Cr-Cu alloy during aging was systematically studied by means of advanced spherical aberration-corrected electron microscopy. Because Cu has a higher atomic number Z than Al, Si, and Mg, it can produce high contrast in the HAADF-STEM mode and is easy to distinguish. Therefore, it is interesting to investigate the morphology and microstructure of the Cu-containing phase and its effects in Al-Si-Mg-Cr alloys. By using atomic resolution electron microscopy, the effect of Cu atoms on the structure of the precipitates is investigated, and the structural characteristics of the nanoscale clusters or precipitates are analyzed.

## 2. Experimental

Two kinds of Al-Si alloys in this work were prepared from commercially pure Al, Mg, Cr, Cu, and Al-Si master alloys. The samples are prepared by conventional die-casting routines. Their chemical compositions are shown in Table 1 below. The samples were heated to 540 °C, held for 8 h, and then quenched in water. Afterward, the samples were aged in an oil bath furnace at various temperatures (160 °C, 175 °C, and 190 °C) for different times followed by air cooling. Hardness measurements were conducted on an HVS-30P digital Micro-Vickers hardness tester. The loading force is 49.8 N and the dwelling time is 15 s. Specimens for TEM observation were electropolished using a twin-jet electropolisher operated at 20 V in a solution of 30 vol.% HNO_3_ and 70 vol.% methanol at temperatures below −25 °C. Then, the electropolished specimens were further thinned by Ar-ion milling using a Gatan precision ion polishing system at 2 kV with an incident beam angle of 2° for 0.5 h. Atomic-resolution high-angle annular dark-field scanning transmission electron microscope (HAADF-STEM) imaging and energy dispersive X-ray spectrometry (EDS) mapping were carried out on a JEM-ARM200F equipped with a cold field emission gun and a probe-Cs corrector. The accelerating voltage was 200 kV. The HAADF-STEM images were acquired with a collection semiangle of 68–280 mrad.

## 3. Results and Discussion

### 3.1. Aging Curves

Figure 1a shows the aging hardness curves of the Al-Si-Mg-Cr alloy at different temperatures (160 °C, 175 °C, and 190 °C). The three aging hardness curves present a similar trend. The aging process can be divided into three stages: underaged, peak-aged, and overaged. With the change in temperature, the aging progress is different. As the aging temperature increases, the time to reach peak hardness gradually decreases, while the peak hardness gradually decreases. Aging hardness curves at 175 °C are used as an example to illustrate the aging progress. The hardness of the alloy after solid solution treatment is approximately 55 HV. At the beginning of aging, the hardness of the alloy increased significantly. However, the hardness growth rate decreases gradually as the aging time increases. The hardness curve reaches its peak after the temperature is held for 9 h and the peak aging hardness is 105 HV. Upon a further increase in aging time, the hardness begins to decrease. This indicates that the alloy begins to enter the overaged stage. When the aging time reaches 16 h, the hardness of the alloy tends to be stable, and the hardness value is approximately 85 HV.

Figure 1b shows the aging hardness curves of the Al-7%Si-0.3%Mg-0.3%Cr-1.5%Cu (wt.%) alloy at different temperatures (160 °C, 175 °C, and 190 °C). By comparing the aging curves of the Al-Si-Mg-Cr-Cu alloy with those of the Al-Si-Mg-Cr alloy, it is found that the aging behavior of the two alloys is similar. However, it can be seen from the figure that the addition of Cu can obviously improve the hardness of the Al-Si-Mg-Cr alloy. The hardness of the solid solution Al-Si-Mg-Cr-Cu alloy is 80 HV, which is approximately 15 HV higher than that of the alloy without Cu. In addition, the peak aging hardness of the alloys is increased by approximately 10 HV by the addition of Cu.

### 3.2. TEM Observation of an Al-Si-Mg-Cr Alloy

For illustrative purposes, the aging process of the alloy at 175 °C is characterized by TEM. As shown in Figure 2a, a large number of nanosized dispersoids can be observed in the solution-treated alloy. In our previous work, we found that most of the dispersoids are$\;\alpha^{\prime }$-${\mathrm{Al}}_{13}{\mathrm{Cr}}_{4}{\mathrm{Si}}_{4}\;$[8]. The solid solution or the supersaturated solid solution may be formed because Mg and Si elements gradually dissolve into the matrix during solid solution treatment. The atomic radius difference between Mg, Si, and Al will cause a large lattice distortion. This kind of distortion can produce a strong stress field, which can bring an obvious strengthening effect to the alloy. Therefore, after solution treatment, the strength of the alloy can be improved to a certain extent, which is mainly from the solution strengthening caused by the supersaturated solid solution and dispersion strengthening caused by dispersoids. Figure 2b shows a TEM image of the alloy after aging for 2 h. It can be seen from the figure that, in addition to the dispersed phase, a high density of dot-like (marked with red circles) and needle-like (marked with yellow rectangles) precipitates are evenly distributed on the matrix. It is obvious that the needle-like precipitate is the $\;\beta^{\prime \prime }$ phase perpendicular to the electron beam, while the dot-like precipitate is the $\;\beta^{\prime \prime }$ phase parallel to the electron beam. Therefore, when the aging time reaches 2 h, the needle-like $\;\beta^{\prime \prime }$ phase begins to precipitate. When the aging time reaches 9 h, the radius of the precipitated phase reaches 3.5 nm and the length of the phase reaches 35 nm, as shown in Figure 2c. In the hardness curve, the alloy enters the peak aging stage. At the same time, the number density of the $\;\beta^{\prime \prime }$ phases further decreased but remained very high. When the aging time reaches 12 h, the sample begins to enter the overaging stage. As shown in Figure 2d, there are a large number of dot-like and needle-like precipitates distributed on the Al matrix. Many large-scale needle-like precipitates occur along ${\left[001\right]}_{\mathrm{Al}}$ and ${\left[010\right]}_{\mathrm{Al}}$.

Figure 3 shows the high-resolution transmission electron microscopy (HRTEM) images of the alloy aged at 175 °C. Figure 3a shows the HRTEM image of a dot-like precipitate in the alloy aged for 2 h and its corresponding fast Fourier transform (FFT) pattern. According to the FFT pattern, the dot-like precipitate is identified as the $\;\beta^{\prime \prime }$-${\mathrm{Mg}}_{5}{\mathrm{Si}}_{6}$ phase. The phase belongs to the monoclinic system, the space group is C2/m, and the crystal parameters are a = 1.51 nm, b = 0.405 nm, c = 0.67 nm, and β = 105° [21,22]. It can be seen in the figure that the $\;\beta^{\prime \prime }$ phase is completely coherent with the Al matrix. The orientation relationship between the $\;\beta^{\prime \prime }$ phases and the matrix is determined as follows:$\;{\left[010\right]}_{\;\beta^{\prime \prime }}//{\left[001\right]}_{\mathrm{Al}},\;{\left(601\right)}_{\;\beta^{\prime \prime }}//{\left(200\right)}_{\mathrm{Al}},\;\mathrm{and}\;{\left(40\bar{3}\right)}_{\;\beta^{\prime \prime }}//{\left(020\right)}_{\mathrm{Al}}.$ After calculation, the interface mismatches between $\;\beta^{\prime \prime }$ and the matrix, ${\left(601\right)}_{\;\beta^{\prime \prime }}//{\left(200\right)}_{\mathrm{Al}}$, and ${\left(40\bar{3}\right)}_{\;\beta^{\prime \prime }}//{\left(020\right)}_{\mathrm{Al}}$, are calculated to be 5% and 6%, respectively. The lattice parameters of nanoscale precipitates $\;\beta^{\prime \prime }$ and the matrix are different. To maintain the coherent interface between the precipitated phase and the matrix, there will be a large lattice distortion and strain field. The strain field can effectively block the movement of dislocations and improve the strength of the alloy. Because a large amount of $\;\beta^{\prime \prime }$ phases precipitate in the matrix, the produced strain field is large enough to produce an effective strengthening effect. Accordingly, the aging curve in Figure 1a shows that the hardness of the alloy was greatly improved during aging for 2 h.

As shown in the HRTEM images in Figure 3b, the larger $\;\beta^{\prime \prime }$ in the peak aging stage is still coherent with the aluminum matrix. The larger the size, the larger the strain field. At this time, the elastic strain field increases to the highest value, which greatly hinders the dislocation movement and produces the maximum strengthening effect, resulting in the highest hardness value.

Figure 3c,d show the HRTEM images of the precipitates aged for 12 h viewed along ${\left[001\right]}_{\mathrm{Al}}$. As shown in Figure 3c, the dot-like precipitates are actually the cross-section morphology of needle-like $\beta^{\prime }$. Metastable $\beta^{\prime }$ belongs to the hexagonal system, the space group is P63/m, and the lattice parameters are as follows: a = 0.705 nm and c = 0.405 nm. Its chemical formula is ${\mathrm{Mg}}_{6}{\mathrm{Mg}}_{3.3}$. From the corresponding FFT image in the figure, we can obtain the orientation relationship between the $\beta^{\prime }$ phase and matrix as follows:$\;{\left[001\right]}_{\;\beta^{\prime }}//{\langle 100\rangle }_{\mathrm{Al},}\;{\left(100\right)}_{\;\beta^{\prime }}//{\left(220\right)}_{\mathrm{Al}}$. The above results indicate that overaging occurs $\beta^{\prime \prime }$→$\beta^{\prime }$. This is the same as the precipitation process of the 6xxx series aluminum alloy [23]. In addition, it also found that a certain amount of rectangular lath phase is precipitated in the overaging stage, as shown in Figure 3d. It was confirmed that these phases are the $B^{\prime }$ phase found in the Al-Mg-Si alloy by Dumolt et al. [24]. The $B^{\prime }$ phase is hexagonal, the space group is P6, and the lattice constants are a = 1.04 nm and c = 0405 nm. From the FFT of the $B^{\prime }$ phase and the surrounding Al matrix, we found that the orientation relationship between the $B^{\prime }$ phase and the matrix is ${\left[001\right]}_{B^{\prime }}//{\langle 100\rangle }_{\mathrm{Al}},\;{\left\{150\right\}}_{B^{\prime }}//{\left(100\right)}_{\mathrm{Al}}$.

The axial direction of the $B^{\prime }$ phase is ${\left[001\right]}_{B^{\prime }}$ The lattice parameter c is the same as the matrix, so the lath-like phase $B^{\prime }$ is coherent with the Al matrix along the long axis. As shown in Figure 3d, interface I between ${\left\{150\right\}}_{B^{\prime }}$ and ${\left(100\right)}_{\mathrm{Al}}$ is marked by the red box. It is found that the mismatch between ${\left\{150\right\}}_{B^{\prime }}$ and ${\left(100\right)}_{\mathrm{Al}}$ is only 0.7%. This shows that the interface in this direction is a coherent interface. At the same time, the yellow box interface II is ${\left\{6\bar{2}0\right\}}_{B^{\prime }}$ and ${\left(010\right)}_{\mathrm{Al}}$. The results show that the interface is a semicoherent interface. It can be inferred that to reduce the interface energy during the process of phase growth interface I of the precipitated $B^{\prime }$ phase will dominate the coarsening process, resulting in the length of interface I being much longer than that of interface II, and $B^{\prime }$ presents a strip shape.

In the overaging stage, the average cross-sectional area of the rod-like $\beta^{\prime }$ and slats $B^{\prime }$ precipitates are more than 20 nm^2^, and the axial length of the precipitates is more than 100 nm. The mismatch degree of these large-scale precipitates with the Al matrix is quite high, which cannot meet the completely coherent interface relationship. Therefore, these precipitates can reduce the lattice distortion and the interface energy with the matrix by maintaining a semicoherent interface relationship with the matrix. However, it also leads to the weakening of the strain field with the matrix. The weakening of the strain field reduces the strengthening effect. Therefore, as the aging curve shows, the hardness of the alloy decreased significantly.

It can be inferred that the precipitation of the $\;\beta^{\prime }$ and $B^{\prime }$ phases are further prolonged with aging time, and $\;\beta^{\prime }$ and $B^{\prime }$ phase transformations will occur to form an equilibrium β phase. The β phase and matrix are noncoherent. The strengthening effect of precipitates will be further reduced. Accordingly, the hardness value is further reduced and tends to be stable.

### 3.3. TEM Observation of an Al-Si-Mg-Cr-Cu Alloy

For comparison, microstructures of the Al-7%Si-0.3%Mg-0.3%Cr-1.5%Cu alloy that were as-quenched, underaged, peak-aged, and overaged were observed by TEM, as shown in Figure 4. In Figure 4a, only the Cr-containing phase and Si particles are found in the solid solution Al-7%Si-0.3%Mg-0.3%Cr-1.5%Cu alloy. At this time, Cu and Mg elements are completely dissolved into the matrix to form a supersaturated solid solution. After cooling in water, a supersaturated solid solution will be obtained, resulting in an obvious solid solution-strengthening effect. Combined with the aging curve, it can be concluded that the addition of Cu improves the hardness of the solid solution alloy, which is the solution strengthening produced by Cu atoms. Figure 4a also shows the interaction of dislocations with dispersoids, where dislocation looping took place. The dislocation-particle interaction has been explained by Safyari et al. [25] that dislocation loops can leave the dispersoids by slip, which is the Orowan mechanism. Figure 4b is the TEM image of the alloy after aging for 6 h. In the figure, it can be seen that many dot-like precipitates appear and are evenly distributed. Figure 4c shows the microstructure of the alloy aged for 12 h. As shown in the figure, a large number of needle-like phases precipitate on the matrix, and the axial direction of the needle-like phase is parallel to ${\langle 100\rangle }_{\mathrm{Al}}$. The axial size of the acicular phase is approximately 35 nm. The aging curve shows that the hardness of the alloy reaches its peak value after aging for 12 h; in other words, the alloy enters the peak aging stage. Figure 4d shows the microstructure of the alloy aged for 16 h. Similarly, many acicular and punctate phases are uniformly distributed on the matrix. The axial direction of acicular precipitates is parallel to ${\left[100\right]}_{\mathrm{Al}}$ or ${\left[010\right]}_{\mathrm{Al}}$. The axial dimension is approximately 150 nm.

To understand the role of Cu atoms during the precipitation of the Al-Si-Mg-Cr alloy, the specimens aged at 175 °C for 1 h, 2 h, 6 h, 12 h, and 16 h are further investigated by the atomic-scale HAADF-STEM.

Figure 5 is a set of HAADF-STEM images of precipitates. It is obvious that the precipitates are actually clusters of specific solute atoms, i.e., the GP zone. It is well known that the intensity in the HAADF-STEM images is approximately proportional to Z^1.7^, where Z is the atomic number. There is no doubt that these bright lines are composed of elements heavier than Al because their contrast is much brighter. According to the later part of the study, these heavier atoms are supposed to be Cu. Two kinds of GP zones are observed after aging for 1 h, as shown in Figure 5a. Figure 5b shows that the two short lines perpendicular to each other present high contrast. It is believed that the Cu atoms are segregated on the interface between the precipitates and the matrix. The typical structures can be classified as type I GP zones. In addition, a relatively long line composed of high-contrast atoms is observed, as shown by the arrow in Figure 5c. It is quite different from the type I GP zone, and this structure is named the type II GP zone. Both the type I GP zone and type II GP zones are completely coherent with the matrix. In fact, there is no structural difference between the GP zones and matrix at this stage. Thus, it is concluded that GP zones formed in the first stage of precipitation have two kinds of structures: one is the perpendicular-line structure (type I GP zone) and the other is the single-line structure (type II GP zone).

Figure 6 shows the HAADF-STEM images obtained from the alloy after aging for 2 h. The difference between the two types of GP zones is more obvious. In Figure 6a, the measured cross-section size of the type I GP zone is approximately 2 nm. The GP zone is completely surrounded by atoms with high contrast. In addition, high-contrast atomic columns are also found inside the GP zone, as marked by the yellow circle in Figure 6a. The corresponding intensity distribution spectrum also shows that these atomic columns have higher intensity, as inserted in Figure 6a. This shows that the Cu atoms not only enriched on the boundary of the GP zone but also diffused to the interior of the GP zone. Figure 6b shows the HAADF-STEM image of the type II GP zone, θ″. The single-line structure has developed into a parallel-line structure. The degree in the figure is divided into three levels: super high contrast, high contrast, and dark contrast, which are shown in white, grey, and black, respectively. The contrast of the precipitates is higher than that of the matrix, which proves that the interface of the precipitates is rich in Cu atoms, but some atomic layers in the axial morphology show higher contrast, which indicates that these atomic layers have heavier atom segregation. The period of this contrast change is one layer in every three layers. The intensity distribution spectrum in the red frame also shows the periodicity and heavy atom segregation in the GP zone.

The contrast analysis and EDS line scan results from A to B are inserted in Figure 6a, from which we can see the enrichment of Si, Mg, and Cu, which shows that the GP zone contains Si, Mg, and Cu elements. Therefore, Cu atoms are mainly concentrated at the interface between the GP zone and matrix, while Si and Mg elements are mainly concentrated in the GP zone. It is well known that the maximum solid solubility of Cu, Mg, and Si in an Al matrix is 5.65%, 1.85%, and 1.65%, respectively. Compared with Cu, the solid solubility of Si and Mg in the Al matrix is much lower. The diffusion rate of Cu atoms in the Al matrix is lower than that of Mg and Si atoms [26,27,28,29]. It is assumed that Si and Mg atoms in supersaturated solid solutions tend to diffuse and aggregate at the beginning of aging and then form atom clusters. Cu atoms form new clusters at the boundary of the formed Si-Mg clusters. At the same time, the relevant research shows that in the Al solid solution, the atomic radius of Mg is 12% larger than that of Al, Si is 6% smaller than that of Al, and Cu is 15% smaller than that of Al [30]. Although the type I GP zone is completely coherent with the matrix, there is a certain elastic strain energy around the GP zone and the matrix due to the difference in the atomic radius between the solute atom and the matrix. The Cu atom with the smallest radius can be segregated at the boundary to minimize this elastic strain energy. Therefore, with increasing aging time, the Cu atom will still gather at the interface between the precipitated phase and the matrix. However, with the coarsening of the precipitates, a small amount of Cu atoms also diffuses into the GP zone, such as place C in Figure 6a.

On the other hand, due to the formation of the GP zone, there is a radius difference between the atoms in the GP zone and the matrix, but to meet the coherent interface relationship between the two phases, lattice distortion will occur at the interface, resulting in an elastic strain field. The strain field can effectively pin dislocations, so it can improve the strength of the alloy. Accordingly, the hardness of the alloy increases.

When the aging time is extended to 6 h, as shown in Figure 7a, the size of the type I GP zone grows further, with a cross-section size of approximately 2.5 nm. Cu atoms are enriched at the interface between the GP zone and matrix and it is found that the content of Cu increases inside the larger GP zone, as shown by the yellow circles in the figure. This shows that with the coarsening of the GP zone, Cu still tends to concentrate at the interface, reducing the overall interface energy. At the same time, the length of the type II GP zone is almost unchanged, while its width has increased, as shown in Figure 7b.

Figure 8 shows the HAADF-STEM images obtained from the alloy after aging for 12 h. The low magnification of the HAADF-STEM image shows that the precipitates or GP zone have similar morphology characteristics at this time, and Cu atoms are segregated at the interface between the precipitated phase and the matrix. However, the high magnification of the HAADF-STEM image in Figure 8a shows that the precipitated area is the $\beta^{\prime \prime }$ phase instead of the GP zone. The addition of Cu atoms does not change the type of precipitate but occupies a specific position of the cell of the precipitate $\beta^{\prime \prime }$. To distinguish it from $\beta^{\prime \prime }$ in the Al-Si-Mg-Cr alloy, we named it $\beta_{\mathrm{Cu}}^{\prime \prime }$. From the Z-contrast and intensity distribution spectrum, it can be seen that the Cu atom tends to not only segregate at the interface but also occupy the position of the Si atom in the $\beta^{\prime \prime }$ phase cell, as shown in the yellow circles in Figure 8a. In addition, the cross-section size of the precipitates exceeded 3 nm, and they were completely coherent with the matrix. The segregation of Cu at the interface lowers the mismatch degree of the two phases. To satisfy the coherent interface relationship, the interface between the $\beta^{\prime \prime }$ phase and the matrix will produce a large lattice distortion and strain field, while the elastic strain field can effectively pin dislocation and improve the strength of the alloy. Therefore, in the peak aging stage, the GP zone has been completely transferred into $\beta_{\mathrm{Cu}}^{\prime \prime }$. A large distortion field is generated at the interface of the phase, which can improve the hardness of the alloy to the greatest extent. At the same time, the width of the type II GP zone is almost unchanged, as shown in Figure 8b.

The HAADF imaging combined with EDS elemental mapping was performed to further identify the precipitates. Figure 9a–f shows a typical HAADF-STEM image of precipitates after aging for 16 h, and the elemental maps were simultaneously recorded. Referring to the TEM results above, it is apparent that the alloy contains needle-shaped phases, as marked by Figure 9a. According to the intensity in the HAADF images, there is no doubt that these phases are composed of elements heavier than Al because their contrast is much brighter. Figure 9b–e shows the EDS maps of Al, Cu, Si, and Mg, respectively. Figure 9b shows that the Al distributes evenly throughout the matrix but diminishes in the phase regions. The needle-shaped phases are rich in Cu and free of Mg and Si. With respect to the HAADF-STEM images and EDS results, these phases are estimated to be $\theta^{\prime }$-Al_2_Cu phases. However, the distribution of Si and Mg exactly coincides with other needle-shaped phases, as shown by the red arrows in Figure 9d,e. They are invisible in the HAADF image due to the very close atomic numbers of Mg (12), Al (13), and Si (14). In the TEM results, these phases are estimated to be $\beta^{\prime }$-Mg_2_Si precipitates. In addition, a very small amount of Cu is found to be enriched in these phases.

The atomic-scale HAADF-STEM was applied to further characterize the phases. Figure 10 shows the enlarged areas from the yellow rectangles (marked by 1,2,3,4) in Figure 9a, respectively. By analyzing the atomic-scale HAADF-STEM images in Figure 10a,b, we can see that this is a $Q^{\prime }\;$phase. The composition of the $Q^{\prime }\;$phase is ${\mathrm{Al}}_{4}{\mathrm{Cu}}_{2}{\mathrm{Mg}}_{8}{\mathrm{Si}}_{7}$. It belongs to the hexagonal system, the space group is P-6, and the lattice parameters are a = 1.03 nm and b = 0.403 nm. The $Q^{\prime }$ phase has a structure similar to that of the $B^{\prime }$ phase. In addition, it can be seen from the figure that there is a triangular substructure in the precipitate, as shown in the yellow triangle in Figure 10a,b. The Cu atom occupies the center of the Mg triangle. This substructure is composed of Mg, Si, and Cu atoms, with Cu atoms as the center and Mg atoms as the framework. Therefore, it can be concluded that the alloy indeed enters the overaging stage. Thus, it is concluded that the addition of Cu inhibits the transformation from the $\;\beta^{\prime \prime }$ phase to the $\beta^{\prime }$ phase and promotes the transformation from the $\;\beta^{\prime \prime }$ phase to the $Q^{\prime }\;$phase. The $Q^{\prime }$ phase structure model shows that Si is arranged in the quadrilateral frame and that Cu atoms are in the interior of the Si frame. It can be inferred that the Cu atom promotes the structural transformation centered on it. In addition, the Cu atom still tends to be enriched at the interface between the $Q^{\prime }$ phase and the matrix, as shown in Figure 10a,b.

Figure 10c,d shows the formation of another precipitate during the overaging process. The precipitate is a lamellar $\;\theta^{\prime }$-$\left({\mathrm{Al}}_{2}\mathrm{Cu}\right)$ phase and evolves from the type II GP zone. It belongs to the tetragonal system, and its thickness direction is [001] and parallel to ${\langle 100\rangle }_{\mathrm{Al}}$. When the thickness direction is perpendicular to the electron beam, the $\;\theta^{\prime }$ phase shows a long axial shape. It can be seen from the figure that the atomic spacing in the thickness direction is approximately 0.29 nm, which is consistent with the theoretical value. The corresponding FFT also proves that it is $\;\theta^{\prime }$. Due to the relatively high content of Cu, the Cu atom promotes the formation of $\;\theta^{\prime }$-$\left({\mathrm{Al}}_{2}\mathrm{Cu}\right)$.

As discussed above, the type I GP zone $\beta_{\mathrm{Cu}}^{\prime \prime }\;$evolved into the $Q^{\prime }$ phase, while the type II GP zone evolved into the $\;\theta^{\prime }$ phase during the aging process. Both the $Q^{\prime }$ phase and the $\;\theta^{\prime }$ phase maintain a semicoherent interface with the matrix. Therefore, the elastic strain field of the interface between the precipitated phase and the matrix decreases, so the hardness of the alloy will decrease at this time. At the same time, when the aging time of the alloy is further extended, the $Q^{\prime }$ phase and the $\;\theta^{\prime }$ phase will be converted into Q and θ. The Q phase and the matrix are semicoherent, but the θ phase and matrix are noncoherent. Therefore, the hardness of the alloy will be further reduced to a stable value.

## 4. Conclusions

In this work, the effect of Cu addition on the precipitation sequence Al-Si-Mg-Cr alloy is studied. The main conclusions are as follows:(1)According to the aging curve at 175 °C, the peak hardness time for the Al-7%Si-0.3%Mg-0.3%Cr alloy is 9 h and the peak hardness value is 105 HV, and the Al-7%Si-0.3%Mg-0.3%Cr-1.5%Cu alloy has values of 12 h and 115 HV, respectively.(2)Two types of GP zones are formed in the first stage of precipitation: one is the type I GP zone and the other is the type II GP zone. The type I GP zone $\beta_{\mathrm{Cu}}^{\prime \prime }\;$ evolved into the $Q^{\prime }$ phase, while the type II GP zone evolved into the  θ ′ phase during the aging process.(3)The aging sequence of the Al-Si-Mg-Cr alloy can be determined as a supersaturated solid solution (SSSS) → GP zones→ $\beta^{\prime \prime }$ → $\beta^{\prime }$/$B^{\prime }$(→β). The aging sequence of the Al-7%Si-0.3%Mg-0.3%Cr-1.5%Cu alloy can be determined as a supersaturated solid solution (SSSS)→GP zone→$\beta_{\mathrm{Cu}}^{\prime \prime }$→$Q^{\prime }\;$+$\;\theta^{\prime }$(→Q + θ).(4)Cu atoms tend to gather at the interface between the precipitates and the matrix during the aging process. Then, Cu atoms diffuse into the precipitates.(5)During the overaging stage, the Cu atom promotes the transition of $\;\beta^{\prime \prime }$→$Q^{\prime }$, which inhibits the transformation of $\;\beta^{\prime \prime }$→$\beta^{\prime }$. At the same time, due to the high content of Cu, Cu atoms and Al atoms formed $\;\theta^{\prime }$.

## Figures and Tables

**Figure 1 materials-15-08221-f001:**
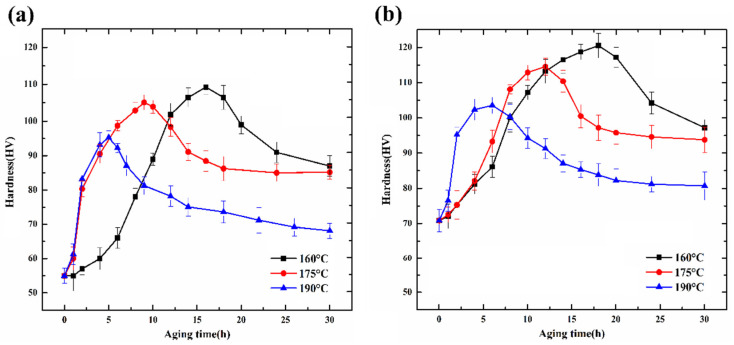
Aging hardness curves of (**a**) Al-7%Si-0.3%Mg-0.3%Cr (wt.%) alloy and (**b**) Al-7%Si-0.3%Mg-0.3%Cr-1.5%Cu (wt.%) alloy under different temperatures.

**Figure 2 materials-15-08221-f002:**
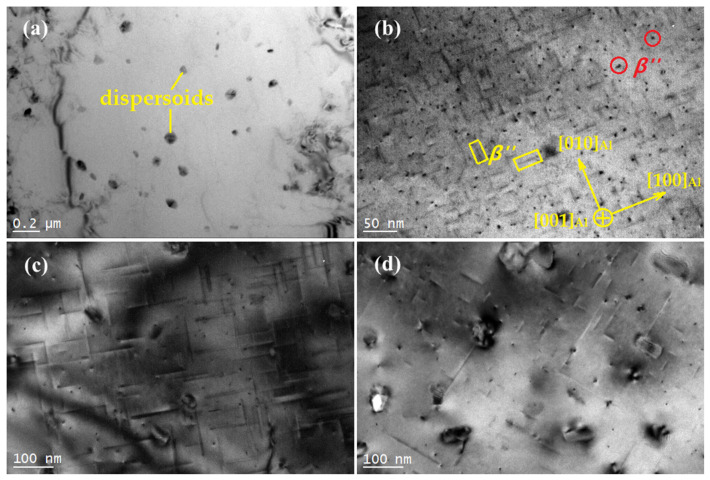
TEM image of the solution-treated and aged alloy,$\;{\left[001\right]}_{\mathrm{Al}}$ (**a**) as solution-treated; (**b**) aged for 2 h; (**c**) aged for 9 h; (**d**) aged for 12 h.

**Figure 3 materials-15-08221-f003:**
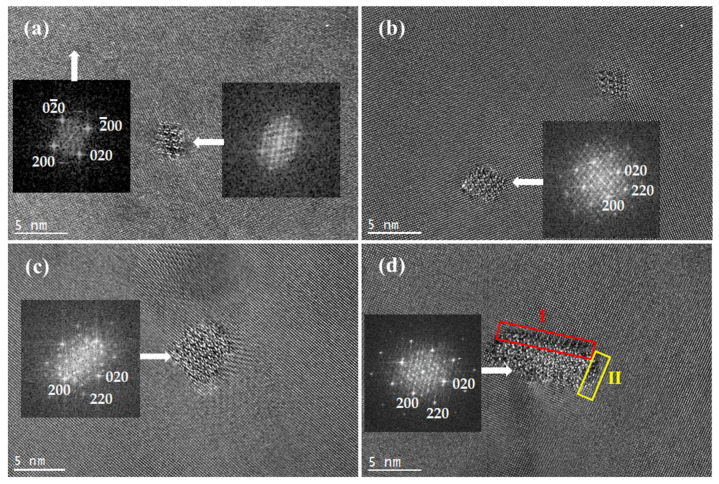
HRTEM image of precipitates and their FFT images,$\;{\left[001\right]}_{\mathrm{Al}}$ (**a**) aged for 2 h; (**b**) aged for 9 h; (**c**) aged for 12 h; (**d**) aged for 12 h.

**Figure 4 materials-15-08221-f004:**
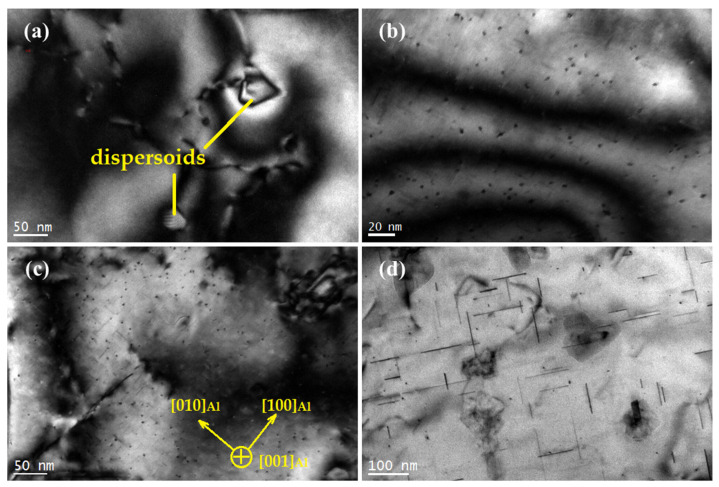
Microstructures of (**a**) as-quenched Al-Si-Mg-Cr-Cu alloy, (**b**) alloy aged for 6 h, (**c**) alloy aged for 12 h, (**d**) alloy aged for 16 h,$\;{\left[001\right]}_{\mathrm{Al}}$.

**Figure 5 materials-15-08221-f005:**
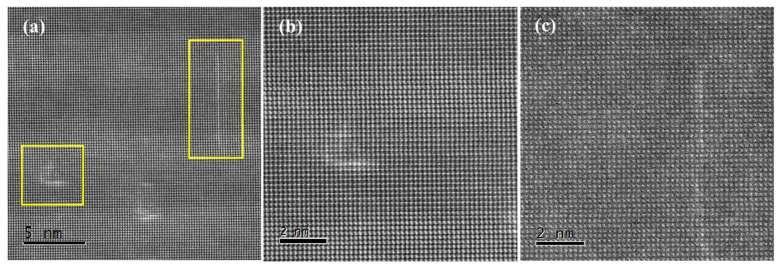
(**a**) HAADF-STEM image of the GP zone after aging for 1 h, and (**b**,**c**) enlarged images of the rectangular areas in (**a**),$\;{\left[001\right]}_{\mathrm{Al}}$.

**Figure 6 materials-15-08221-f006:**
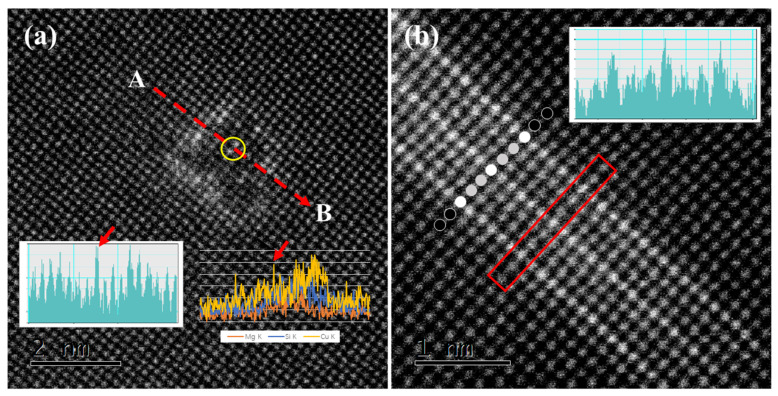
HAADF-STEM image of the GP zone after aging for 2 h,$\;{\left[100\right]}_{\mathrm{Al}}$ (**a**) Type I GP zone and the inserted EDS line scan results; (**b**) Type II GP zone.

**Figure 7 materials-15-08221-f007:**
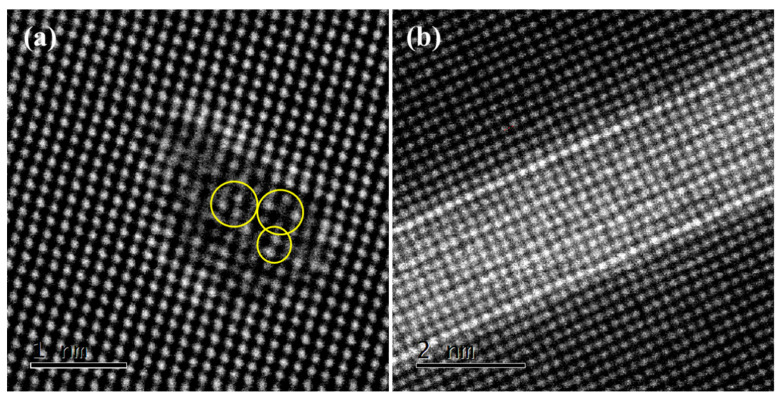
HAADF-STEM image of the GP zone in the alloy aged for 6 h,$\;{\left[100\right]}_{\mathrm{Al}}$ (**a**) type I GP zone; (**b**) type II GP zone.

**Figure 8 materials-15-08221-f008:**
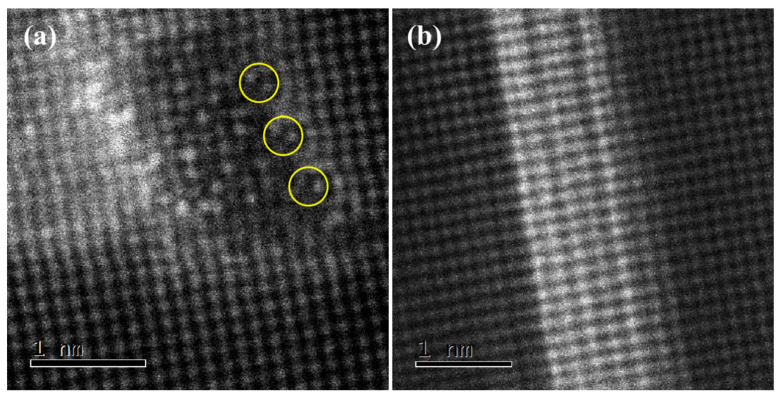
HAADF-STEM images of the precipitate after aging for 12 h.$\;{\left[100\right]}_{\mathrm{Al}}$ (**a**) type I GP zone; (**b**) type II GP zone.

**Figure 9 materials-15-08221-f009:**
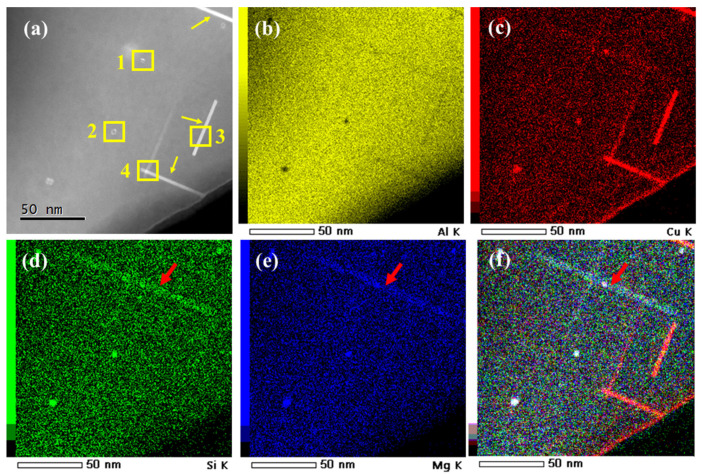
HAADF-STEM images of Al-7%Si-0.3%Mg-0.3%Cr-1.5%Cu alloy (**a**), corresponding EDS maps of elements Al (**b**), Cu (**c**), Si (**d**) and Mg (**e**), and their overlay image (**f**).

**Figure 10 materials-15-08221-f010:**
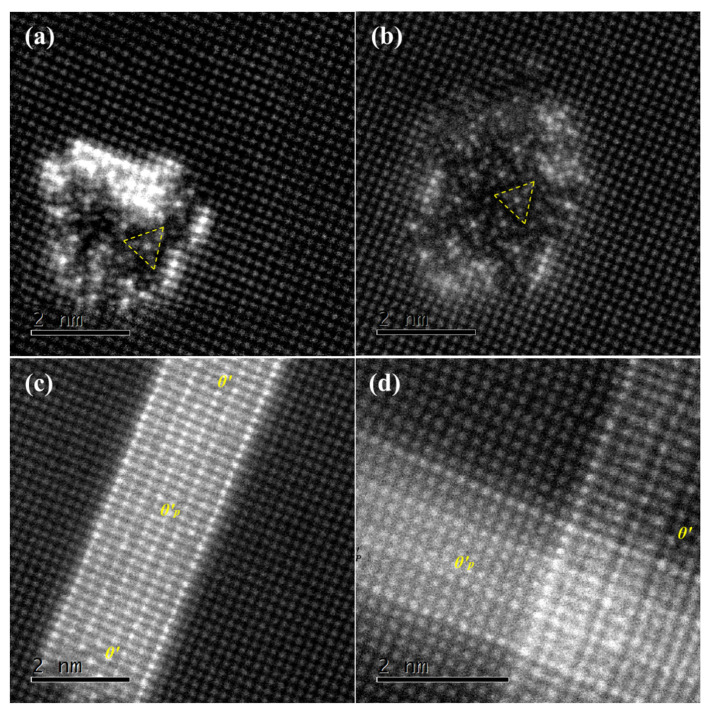
HAADF-STEM image of the precipitate after aging for 16 h,$\;{\left[001\right]}_{\mathrm{Al}}$ (**a**) $Q^{\prime }\;$phase; (**b**) $Q^{\prime }$ phase; (**c**) $\theta^{\prime }$ phase; (**d**) $\theta^{\prime }$ phase.

**Table 1 materials-15-08221-t001:** Composition of cast aluminum alloys (wt.%).

Alloy	Si	Mg	Cr	Cu	Fe	Al
Al-Si-Mg-Cr	7	0.3	0.3	0	<0.15	Bal.
Al-Si-Mg-Cr-Cu	7	0.3	0.3	1.5	<0.15	Bal.

## Data Availability

Not applicable.

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
