# Peer review of "The Effect of Cu Addition on the Precipitation Sequence in the Al-Si-Mg-Cr Alloy"

_materials, 2022, doi:10.3390/ma15228221_

Round 1

Reviewer 1 Report

The article „The effect of Cu addition on the precipitation sequence in the Al-Si-Mg-Cr alloy“ is written at a high level and the authors used appropriate analytical techniques. The article provides original and clear results for a comparison of aging processes in Al-Mg-Si-Cr alloy and Al-Mg-Si-Cr-Cu alloy with 1.5 wt% Cu.

I recommend the article for publication after following minor modifications:

- it would be appropriate to explain the designation β-〖Mg〗_2 Si. How does it differ from β-Mg2Si? The phase-designation should be unified.

- add subscripts to chemical formulas, especially to the chemical notation of precipitates, for example use θ-Al2Cu instead of θ-Al2Cu

- hardness values should be designated consistently with the letters HV, for example 85 HV, not 85 hv.

- when describing the Figure 2 there are results stated in the text, such as:

„They are α′-??13??4??4, a small amount of Si particles and α-??12(??, ??)3??2.“ or „Many large-scale needle-like precipitates occur along [001]?? and [010]??.“

It is not possible to determine these results from the figure alone and they should be supported by results from electron diffraction, chemical analysis or atomic resolution. The same is true for the Figure 4. It should be stated how the information was obtained.

 - the space group designation should be unified, for example, instead of „p 63/m“, „P-6“... use P63/m, P6...

- explain the difference between designations β_Cu^'' and ???′′

- correct minor typos

Reviewer 2 Report

In the manuscript, the effect of Cu additions and heat treatment on the precipitation sequence of the Al-Si-Mg-Cr alloy was systematically studied using by means of advanced spherical aberration-corrected electron microscopy. The manuscript is well written, the results are well presented and discussed in detail by the authors. The obtained in this work results will certainly be of interest to the scientific community in this area.

I have the following comments on the manuscript:

·  Abstract. The abstract contains many small blots like «β_Cu^''». Blots need correction.

·       Line 30. The β^'' and β^' must be corrected to β'' and β'.

·       Experimental. Was the chemical composition controlled by the authors?

·       Line 90. The “hv” must be corrected to ”HV.

·       Line 93. The “hv” must be corrected to ”HV.

·       Line 96. The “hv” must be corrected to ”HV.

·       Figure 1. The authors need to make Figure 1 more visual. For example, compare alloys with copper and without copper on one graph. Or make the Y-axis values the same.

·       Figure 2. The authors describe the structural components in the TEM-images in great detail. The identification of structural components in the images would make the manuscript more convenient and scientifically significant.

·       Figure 3. Authors must index FFT patterns for HRTEM images.

·       Figure 4. The identification of structural components in the images would make the manuscript more convenient and scientifically significant.

·       References. I think the review of publications on the subject of investigation can be expanded. I think there are many works in recent years on the study of precipitates in Al-Si-Mg, Al-Mg-Cu, Al-Mg-Cu-Cr and Al-Si-Mg-Cu alloys.

Reviewer 3 Report

The manuscript is about the effect of Cu addition on the precipitation sequence in the  Al-Si-Mg-Cr alloy, the following corrections must be done before publication:

1. Some EDS maps should be added to Fig. 2 to identify the chemical composition of the particles. 

2. The inverse FFT should be added to Fig. 3. 

3. Fig. 4 shows the interaction of the dislocations with particles. This should be explained using the following paper

https://doi.org/10.1016/j.corsci.2021.109895

4. The effect of the particles on mechanical properties should be explained in detail.

Round 2

Reviewer 2 Report

The authors corrected and significantly improved the manuscript. I think the manuscript can be accepted for publication.

Reviewer 3 Report

The manuscript is acceptable.